# Bearing Fault Reconstruction Diagnosis Method Based on ResNet-152 with Multi-Scale Stacked Receptive Field

**DOI:** 10.3390/s22051705

**Published:** 2022-02-22

**Authors:** Hu Yu, Xiaodong Miao, Hua Wang

**Affiliations:** School of Mechanical and Power Engineering, Nanjing Tech University, Nanjing 211816, China; 202061207070@njtech.edu.cn (H.Y.); wanghua@njtech.edu.cn (H.W.)

**Keywords:** subway train, fault diagnosis, data reconstruction, deep learning, residual neural network, Gramian angular field

## Abstract

The axle box in the bogie system of subway trains is a key component connecting primary damper and the axle. In order to extract deep features and large-scale fault features for rapid diagnosis, a novel fault reconstruction characteristics classification method based on deep residual network with a multi-scale stacked receptive field for rolling bearings of a subway train axle box is proposed. Firstly, multi-layer stacked convolutional kernels and methods to insert them into ultra-deep residual networks are developed. Then, the original vibration signals of four fault characteristics acquired are reconstructed with a Gramian angular summation field and trainable large-scale 2D time-series images are obtained. In the end, the experimental results show that ResNet-152-MSRF has a low complexity of network structure, less trainable parameters than general convolutional neural networks, and no significant increase in network parameters and calculation time after embedding multi-layer stacked convolutional kernels. Moreover, there is a significant improvement in accuracy compared to lower depths, and a slight improvement in accuracy compared to networks than unembedded multi-layer stacked convolutional kernels.

## 1. Introduction

Subway trains are integral to traffic systems, modernization, and urban culture [1,2,3]. However, because these axle boxes of a subway train support the whole weight of the subway vehicle and ensure the reliability of a subway train [4,5], and the rolling bearings are the vitally important component to transfer loads and torque through which are filtered by an air spring to shaft. Hence, failures unavoidably occur in rolling bearings and result in economic loss or even human casualties. As a result, fast and accurate fault diagnosis of axle box bearings can be used to maintain the smooth operation of urban rail transit and extend service time as well as ensure travel safety.

The fault diagnosis methods used mainly for rolling bearings can be classified into two categories, vibration-based signal analysis and machine-learning-powered methods [6,7,8,9,10]. In general, vibration-based signal analysis methods detect faults by extracting fault-related vibration components and characteristic frequency. However, vehicle-mounted sensors used to extract other irrelevant vibration signals including the shaft and gearbox, etc., when subway trains operate at high speeds. Hence, in the early stage of faults, bearing-related fault signals used to be overwhelmed by overstated other components and harmonics or environment noise. Therefore, it is hard to extract pure fault-related vibration signal by traditional vibration-based signal analysis.

Machine-leaning-powered fault diagnosis methods detect faults by extracting a series of statistical parameters (e.g., such as kurtosis, root mean square, energy and entropy.) to represent bearings’ health states. Meanwhile, these parameters can be used to train classifiers (e.g., such as a support vector machine (SVM), a deep neural network (DNN), or a Bayes network) to classify different fault characteristics. Among them, SVM is a class of generalized linear classifiers that perform binary classification of data in a supervised learning manner, neural networks are based on extensions of perceptrons, while DNN can be understood as neural networks with many hidden layers. Nevertheless, the extracted statistical parameters cannot ensure the accuracy of distinguishing different faults. Therefore, finding suitable training parameters to train traditional classifiers is a long-term challenge for machine-learning-powered fault diagnosis methods [11].

In recent years, deep learning (DL) methods, which take vibration-related signals as input data, has been applied in various fields [12,13,14,15,16]. For example, S. Roy et al. [12] applied the successful application of DL in medical imaging to COVID-19 as well as paved the way to future research on DL for the assisted diagnosis of COVID-19 from medicine imaging datasets. K.B. Lee et al. and H. S. DIKBAYIR et al. [13,14,15] detected vehicles in different complex driving environments based on DL. For fault-related signal features extraction, traditional machine-learning-powered methods, which rely on fault-related preprocessing, are the lack of multiple levels of nonlinear transformations [17]. DL cannot only adaptively extract deep features of fault characteristics from anc input layer but can also ease the difficulty of parameters optimizations.

Among the DL theoretical methods, diagnostic models with classical convolutional neural network (CNN) structure are the most widely used, such as Alex Net, VGG, etc. However, because of the big data with large volume, vary modalities, fast generation and large value but low density, these network models have no choice but to improve the depth to parse the massive data, so it will cause huge training parameters and overfitting. By contrast, deep residual networks (ResNets) are an effective variant of CNNs, which can use identity shortcuts to ease the difficulty of parameters optimization [18,19,20,21]. ResNets and their variants have applied for fault diagnosis in a few papers [22,23,24,25,26,27,28]. For instance, C. Zhou et al. [24] analyzed the COVID-19 chest X-raay images based on image regrouping and ResNet-SVM. As a consequence, this method can reach 93% accuracy on a relatively small dataset. M. Zhao et al. [27] proposed deep residual shrinkage network (DRSN), which is an evolution of ResNets. Compared with structure of ResNets, DRSN has a shrinkage block (soft threshold function) and show that the method is effective for high noise fault diagnosis.

The developed residual neural networks can adapt to large-scale data and have good nonlinear expression capability, but a large number of research is based on 1D fault vibration signals, which makes full use of the self-extraction capability of DL but also limits the diagnostic accuracy. This article develops a fault reconstruction characteristics classification method using ResNet-152 inserted with multi-layers convolutional kernels. The data is structured by convolutional units with multi-layer stacked convolutional kernels to enhance the nonlinear representation. One-dimensional vibration signals processed by Gramian angular summation field (GASF) are easier to manipulate by convolutional layers. The main contributions of this paper are as follows:(1)Three-layers stacked convolutional kernels are inserted into ultra-deep ResNets to replace large-size or less-layers convolutional kernels to improve the nonlinear representation of feature images;(2)The fault datasets are reconstructed to increase the data scale and retain the temporal features in the fault data, while reducing the difficulty of the convolution process;(3)Research on axle box bearings for subway trains to improve the efficiency and accuracy of diagnosis of this component.

Additionally, this study verifies the role of superimposed convolution kernels in specific objects for the first time on the basis of the theory; another novelty is the training of deep learning networks using reconstructed fault feature signals, as researchers have overlooked the importance of modest feature engineering while focusing too much on the powerful learning ability of deep learning.

This paper is organized as follows: in Section 2, ResNets-related methods are introduced. In Section 3, details of the design of fundamental architectures for ResNet-152-MSRF, data reconstruction methods, experimental protocols and complete experimental results are presented. In Section 4, the experimental results are summarized, and the advantages and disadvantages of each model are sorted out.

## 2. Basic Components

### 2.1. Basic Structure of Residual Neural Networks

ResNets share many of the same components as traditional CNNs, such as convolution layers, rectifier linear unit (ReLU), activation function, batch normalization (BN), loss function and pooling layers et al. In the fact, the pooling layer, which downsamples fault-related features to submit to the next block, can be used or not in many deep neural networks. The theories of these basic components are described as follows.

The convolution operation in a CNN is the key component of the entire network and is the essential difference from a fully connected (FC) neural network. The convolutional layer in a CNN can effectively reduce the amount of the trainable parameters, so that the training speed of the model is greatly improved. In a neural network, the fewer the trainable parameters, the less likely the network will be over-fitted. The formula for the convolution operation can be expressed as follows:(1)$$z_{u,v}^{\left(l\right)}=\sum_{i=-\infty }^{\infty }\sum_{j=-\infty }^{\infty }x_{i+u,j+v}^{\left(l-1\right)}\cdot k_{rot}{}_{i,j}^{\left(l\right)}\cdot X(i,j)+b^{\left(l\right)}$$where $x_{i+u,j+v}^{\left(l-1\right)}$ is the (l−1)th channel of the input feature map, $k_{roti,j}^{\left(l\right)}$ is the lth corresponding convolution kernel, X(i,j) is the activation function of the corresponding layer, $b^{\left(l\right)}$ is the corresponding bias term, i,j are the relative positions during feature mapping, $z_{u,v}^{\left(l\right)}$ is the lth channel of output feature map. The convolution operation can be repeated several times to obtain a large number of feature maps.

BN is an important technique for normalizing feature data as a trainable process to be inserted into the deep learning architecture [29,30]. Deep networks training is a complex process, whenever a small change occurs in the first few layers of the network, then the later layers will be cumulatively amplified down. Hence, the purpose of BN is to reduce the internal covariate shift, in which updates of the front layer training parameters will lead to changes in the distribution of the back layer input data. As a matter of fact, BN force the distribution of the input value of any neuron in each layer of the neural network back to a standard normal distribution with a mean of zero and a variance of one, so that the activation input value falls in the region where the nonlinear function is more sensitive to the input. BN operation is expressed as follows:(2)$$\begin{array}{l} \mu_{\beta }=\frac{1}{m}\sum_{i=1}^{m}x_{i}\\ \sigma_{\beta }^{2}=\frac{1}{m}\sum_{i=1}^{m}{\left(x_{i}-\mu_{\beta }\right)}^{2}\\ {\hat{x}}_{i}=\frac{x_{i}-\mu_{\beta }}{\sqrt{\sigma_{\beta }^{2}+\varepsilon }}\\ y_{i}=\gamma {\hat{x}}_{i}+\beta \to BN_{\gamma ,\beta }(x_{i}) \end{array}$$where $x_{i}$ and $y_{i}$ represent the input and output feature of the ith observation in a mini-batch. γ and β are two trainable parameters to adjust the distribution. ε is a constant that tends to zero.

Loss function is used to measure the quality of a set of parameters by comparing the difference between the expected output and the true output. In multi-category tasks, the cross-entropy error used to be the objected function to be minimized. Compared with other traditional error functions, cross-entropy can promise a higher training efficiency. Apart from that, in order to strengthen the feature, cross-entropy is usually used with the softmax function to map the output from zero to one. Then softmax function can be expressed as follows:(3)$$\begin{array}{l} p(y|x)=\frac{e^{\left(W_{y}.x\right)}}{\sum_{c=1}^{C}e^{W_{c}.x}}\leftarrow W_{y}.x=\sum_{i=1}^{d}W_{yi}.x_{i}=f_{y}\\ ⇓\\ p(y|x)=\frac{e^{\left(f_{y}\right)}}{\sum_{c=1}^{C}e^{\left(f_{c}\right)}}\to soft\max{\left(f\right)}_{y} \end{array}$$
the first step is to take the yth row of W and multiply that row with x as well as compute for all $f_{c}$ for c=1,…,C, and then apply the softmax function to get a normalized probability. Cross-entropy is expressed as follows:(4)$$H(p,q)=-\sum_{i=1}^{n}p(x_{i})\log q(x_{i})$$
where $p(x_{i})$ is the ith actual probability of observation. After calculating the cross-entropy error, the gradient descent algorithm is used to optimize the parameters, and then the network is fully trained after several iterations.

### 2.2. Insertion of Multi-Scale Superimposed Receptive Field

In this section, the motivation for fault characteristics reconstruction and multi-scale superimposed receptive field that insert into the architecture of deep residual network are introduced.

Receptive field is the convolutional kernel which realize the local perception of the corresponding input, the implementation is a weighted summation over a local region of the input. The size of the convolution kernel must be larger than 1 to have the effect of enhancing the perceptual field, so that the most commonly used convolutional kernel for feature extraction cannot be 1. Convolution kernels of even size cannot guarantee that the input feature map size and output feature map size remain unchanged even if padding is added symmetrically (e.g., if the input is 4 × 4 and the convolution kernel size is 2 × 2 and padding is 1 on each side, there will be a total of 5 outputs after sliding, which will not correspond to the input). Compared with bigger convolution kernels, multi-layer stacked small-sized convolution kernels have more activation functions, richer features and greater discernment. Convolution operation is accompanied by an activation function, and the use of more convolution kernels can make the decision function more discriminative.

Multi-layer stacked convolutional kernel replacement for large size convolution kernel involves parameter calculation. Table 1 shows the comparison of whether stacked convolution is used or not for different kinds of networks or for the same kind of networks with different depths, including VGG-16, VGG-19, ResNet-50, ResNet-152. As shown in Table 1, multi-layer stacked convolutional kernels have a larger number of parameters compared to large size convolutional kernels and few-layer convolutional kernels, but parameters growth rates are all stable at less than 1%, the replacement of 7 × 7 convolutional kernels with 3 × 3 + 3 × 3 + 3 × 3 stacked convolutional kernels in the VGG-19 network has the smallest parameter growth rate of 0.06%, and the replacement of 5 × 5 convolutional kernels with 3 × 3 + 3 × 3 stacked convolutional kernels in the ResNet-152 network has the largest parameter growth rate of 0.9%.

The superiority of the residual network can also be seen in Table 1, where the number of trainable parameters of the ResNet with a depth of 152 layers is less than 20% of that of the VGG network with only 16 or 19 layers, so that ResNets are well suited for embedding stacked convolutional kernels, which stabilize the number of parameters while keeping the network lightweight.

## 3. Design of Fundamental Architectures for ResNet-152-MSRF

In this section, the architecture of ResNet-152-MSRF are elaborated.

The core part of the developed ResNet-152-MSRF is shown in Figure 1, the convolution process is achieved by stacked convolutional kernels. The input image is output to the next stage with tensor data with deep nonlinear features by the action of a three-layer stacked convolution kernel. Neural networks gradually lose local features at each layer through pooling as the depth increases, which is fatal for fault diagnosis. ResNets, on the other hand, lead to ultra-deep network structures. ResNet-152-MSRF has 152 convolutional layers, and each convolutional layer performs 3 nonlinear transformations because of the embedded 3-layer stacked convolutional kernel, and then average pooling is used between each layer as well as the number of feature channels increases and the feature size decreases. In Figure 2 is the overall architecture of ResNet-152-MSRF. The input features of the previous convolutional layer are added to the output features by identity shortcutting, prerequisite is to ensure the same shape (e.g., the input shape of the previous layer is 64 × 64 × 16, then the output feature shape is the same). Dropout(0.5) function is used between each convolutional layer to randomly reduce the number of neurons to prevent overfitting, 0.5 represents the random neuron discard rate. Finally, the fully connected layer is connected, and the number of output nodes is the same as the number of categories.

The advantage of using this architecture is that it can train large datasets well, and the network is deep enough to make sufficient nonlinear transformations to allow the computer to discriminate features.

## 4. Experiment Results

### 4.1. Data Collection and Processing

The drive-train dynamic simulator used for this experimental data acquisition is shown in Figure 3. The entire device consists of a motor, an acceleration sensor, a magnetic brake and a subway train axle box. The load and speed conditions of the bearings in the axle box are determined by the motor, and the sampling frequency is determined by the acceleration sensor. The acceleration sensor is installed between the motor and the axle box (i.e., the input of torque) of the axle box, raw vibration data were collected for 4 health conditions with a sampling frequency of 12,000 Hz. As shown in Table 2, the four health conditions are summarized, including one healthy condition and three faults. The bearings under each health condition are subjected to different loads of 36 KN and 72 KN, respectively.

The data processing after the raw signal acquisition is shown in Figure 4. The raw vibration signal collected by the acceleration sensor is sampled to more than 120,000 points per series, The number of sampling points per series for each of the four health conditions is shown in Table 2. By data splitting, the normalized vibration signal is split into several signal fragments, here the size of each segment is set to 800, which can effectively ensure the sparsity of the features and the size of the dataset, and then repeat the operation to complete the splitting of all the data, import the completed splitting data into the GASF for data reconstruction as well as divide the obtained 2D time-series graph into training set and test set, the specific number of samples and testing set samples are shown in Table 3.

Although deep learning is well developed for computer vision, it is difficult to build predictive models when encountering time series. For different raw input data such as 1D vibration signal data, it is necessary to reconstruct the signals of 1D time series into the form of 2D images, thus the advantages of computer vision can be fully exploited. The 2D timing diagrams of the four health conditions converted from 1D vibration signals by GASF are shown in Figure 5.

### 4.2. Hyperparameters Setup

The four neural networks covered in this paper, including the two residual networks with embedded stacked convolutional kernels, are all implemented based on the tensorflow2.0 framework. Experiments were conducted on a thinkstation with an Intel Xeon W-2223 CPU and an NVIDIA Quadro P220 GPU. In this section, the initialization and setting of hyperparameters are described in detail.

Because there is no clear consensus on the hyperparameter settings for classical CNNs and ResNets [27]. The genetic algorithm (GA) is advanced and outstanding; however, there are hyperparametric results of the optimization of the algorithm itself, which need to be verified experimentally in the next stage, so it is not intuitive. Hence, in this paper, the hyperparameters are set according to the experiment results shown in Figure 6. The experiments are conducted based on the representative VGG-16 and ResNet-152 built in this paper and have shown that when the activation function of the convolution layer is set to ReLU, it converges faster and with higher accuracy than tanh. Moreover, when the loss function is set to mean square error (MSE), the diagnostic accuracy is significantly inferior to that of cross entropy. The effect of the Adam optimizer is similar to RMSprop, so this paper chooses the more commonly used Adam optimizer. In addition, other relevant parameters are shown in Table 4. The input data shape set to 128 × 128 and the depth to three. The meaning of Conv_2 in Table 4 is the number of 2D convolutional layers, activation function_1 refers to the activation function used in the convolutional layer, the ReLU function ensures the sparsity of the network and reduces the interdependence of the parameters, alleviating the overfitting. RBU refers to whether identity shortcut mapping is used in the network structure, where 1 means used and 0 means not used.

Activation function_2 represents the FC activation function. Softmax function is suitable for multi-classification tasks. The learning rate is also uniformly set to 0.0001, which is kept as the same as classical ResNets [18]. The use of the FC loss function has been explained in Equation (3) and the number of output nodes in the FC is set to 4, which is the same as the number of health conditions and contains one normal condition and 3 fault states. The batch size has been set to 20.

### 4.3. Comparison of Six Target Diagnostic Models

As shown in Table 3, the total sample size reached 21,112, in addition to the number of feature pixel points which reached 345,899,008. Compared with the original data scale, the data scale has increased approximately 42.88 times, so experiments were conducted under a scheme of sixfold cross validation. The total dataset is divided into six subsets, five of them are used as the training set and one as the test set, and then the average accuracy and loss are taken as the experimental results after cross validation. In addition, this experiment is conducted for different classification tasks, and three sets of experiments are conducted to verify the diagnostic performance of the model under different classification tasks.

The general operation route of the experiment is shown in Figure 7, including the basic principles of GASF and architecture of residual building block (RBB), in which RBB-1 maps the input directly to the output and RBB-2 deals with the input by convolution and BN.

Comparison of the average accuracy of the six neural network models trained and validated on four different health condition datasets is shown in Figure 8, and the average accuracies on the validation sets for all cases are given in Table 5. ResNets are significantly superior to the general CNN in diagnostic performance in terms of network structure alone. Moreover, for large-scale input data with deep features, residual networks show an extraordinary learning ability. The residual network achieves good accuracy in the more difficult quadruple classification task, to over 95%, and is relatively stable in the simpler quadruple classification task, with fluctuations below 0.5%.

The average calculation time of six different neural network models is shown in Table 6, where the validation batch is set to 10, and that on the training set is set to 20, as a result, ResNet-152-MSRF takes more time than others. Obviously, comparing the number of trainable participants of the network in Table 1, it is clear that it is related to the size of the number of parameters. It is worth noting that both ResNet-50 and ResNet-152 with stacked convolutional kernels embedded (i.e., ResNet-50-MSRF and ResNet-152-MSRF) saw their computation times rise by only 8.201% and 5.577%, respectively.

### 4.4. Comparison between ResNet-152-MSRF and ResNet-50-MSRF

As shown in Table 5, the average diagnostic accuracy of ResNet-152-MSRF is 2.67% higher than that of ResNet-50-MSRF in the four-classification task, 2.28% higher in the three-classification task, and 0.62% higher in the two-classification task.

K. He et al. verified that neural networks lead to lower accuracy when the depth is significantly increased [18]. However, as shown in Figure 9, ResNet-152-MSRF is more stable than the ResNet-50-MSRF, with no significant fluctuations in diagnostic accuracy or error. In contrast with Figure 9a,c. ResNet-50-MSRF achieves faster accuracy and error convergence than ResNet-152-MSRF on the training set, reaching more than 80% training accuracy after 15 iterations, and then combined with the calculation time comparison in Table 6.

## 5. Conclusions

Rapid diagnosis of bearings inside axle boxes of subway trains is an important task to accelerate the modular maintenance of the whole train. This article mainly develops a fault diagnosis method based on ResNet-152-MSRF. In addition, the traditional 1D fault signal is reconstructed using GASF, so that 2D convolution layers can be built to facilitate feature extraction.

(1)Evidenced by the experiments, ResNet-50 and ResNet-152 improved by 24.99%, 37.69%, 26.13% and 38.83% relative to VGG-16 and VGG-19, respectively. Additionally, the result indicates that networks with RBB are more suitable for large-scale deep feature extraction;(2)Evidenced by the data reconstruction, the scale of the obtained data is increased by about 42.88 times compared to the previous 1D time series signal, which is effective for data enhancement;(3)By embedding a multi-layered receptive field, the developed ReNet-152-MSRF enhances the accuracy by 9.07% compared to ResNet-152, and time cost increases non-significantly. ResNet-152-MSRF has a 2.67% and 1.87% higher average diagnostic accuracy, respectively, than ResNet-50-MSRF in different tasks. This suggests that deeper networks do not necessarily affect accuracy and perform well when trained on reconstructed fault data.

In the future, the method proposed in this paper will be verified in more real-time monitoring platform and more accurate models will be obtained.

## Figures and Tables

**Figure 1 sensors-22-01705-f001:**
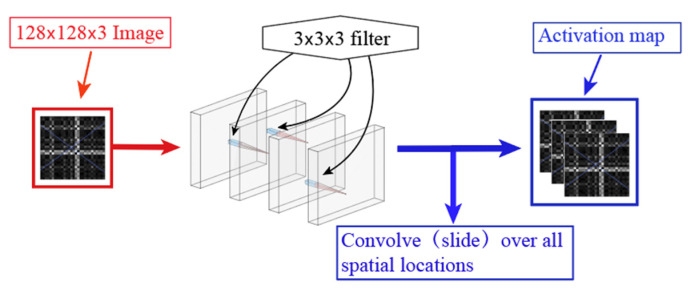
Convolutional process of multi-layer convolutional kernel embedded in residual network.

**Figure 2 sensors-22-01705-f002:**
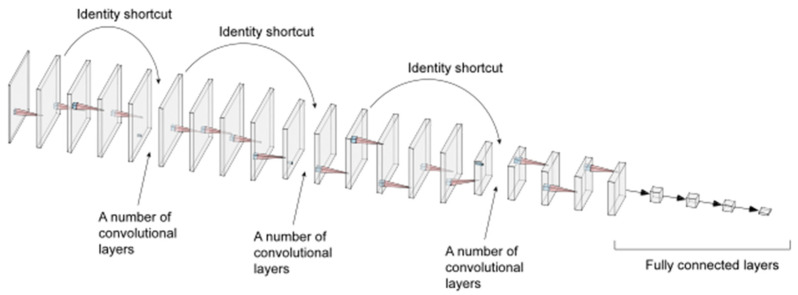
Overall architecture of ResNet-152-MSRF.

**Figure 3 sensors-22-01705-f003:**
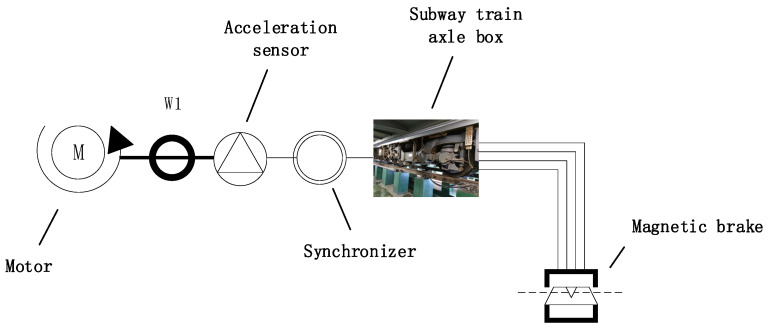
Drive-train dynamic simulator for data collection.

**Figure 4 sensors-22-01705-f004:**
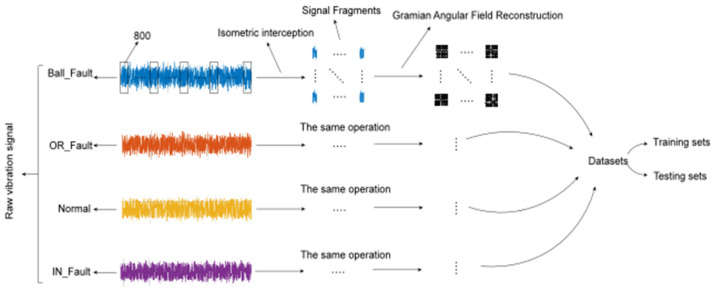
General illustrator of data processing.

**Figure 5 sensors-22-01705-f005:**
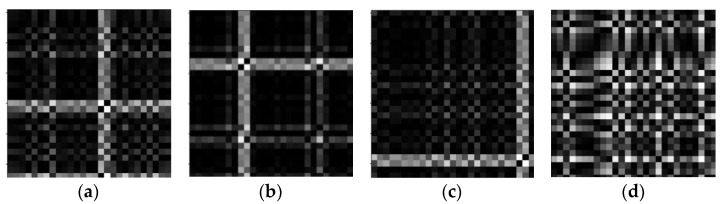
Two-dimensional fault timing diagram obtained from a 1D vibration signal using GASF. (**a**) Roller fault, (**b**) inner raceway fault, (**c**) outer raceway fault and (**d**) normal.

**Figure 6 sensors-22-01705-f006:**
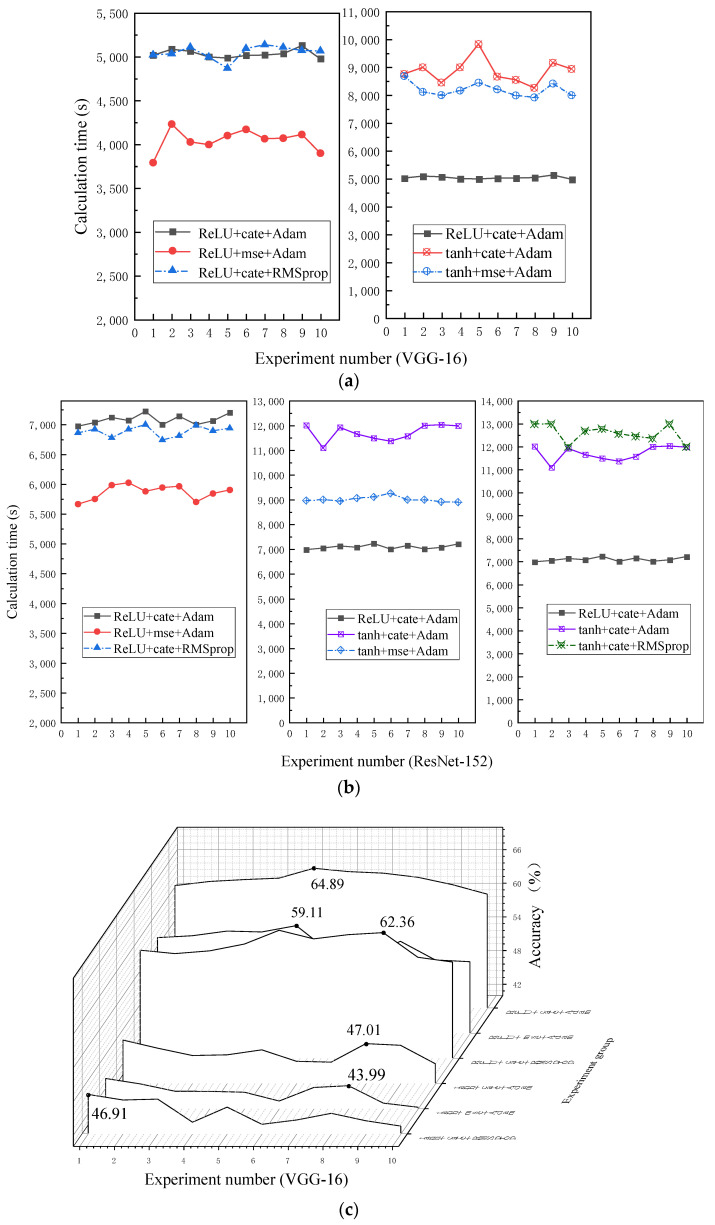
Effects of several hyperparameter settings in VGG-16 and ResNet-152. (**a**) Calculation time of VGG-16, (**b**) calculation time of ResNet-152, (**c**) accuracy comparison of VGG-16, (**d**) accuracy comparison of ResNet-152.

**Figure 7 sensors-22-01705-f007:**
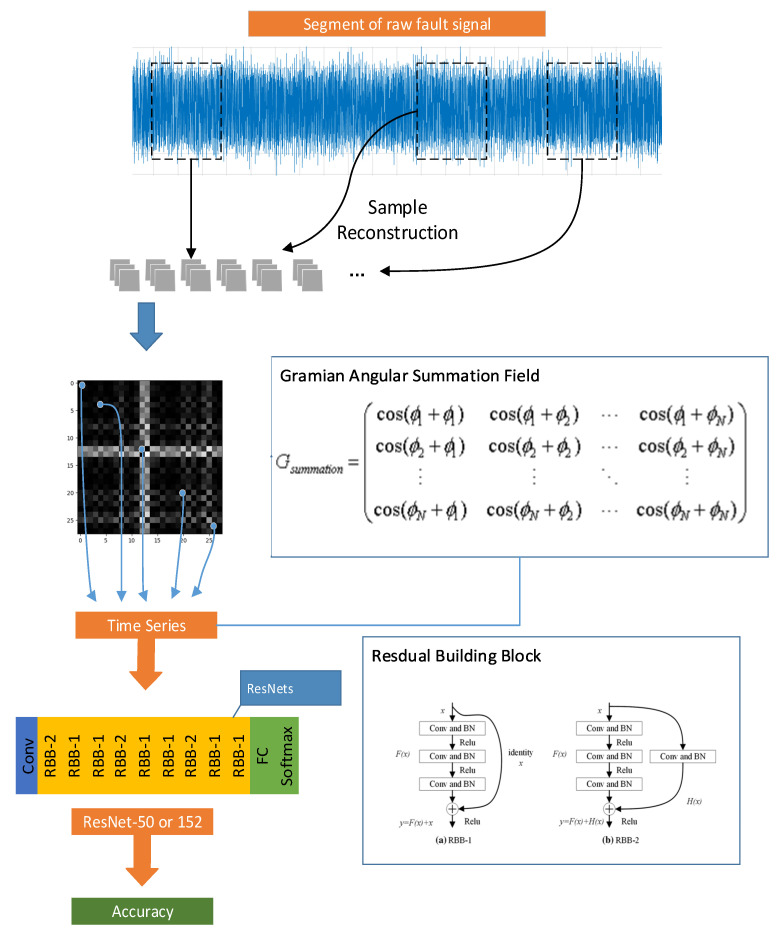
The illustration of research operation route.

**Figure 8 sensors-22-01705-f008:**
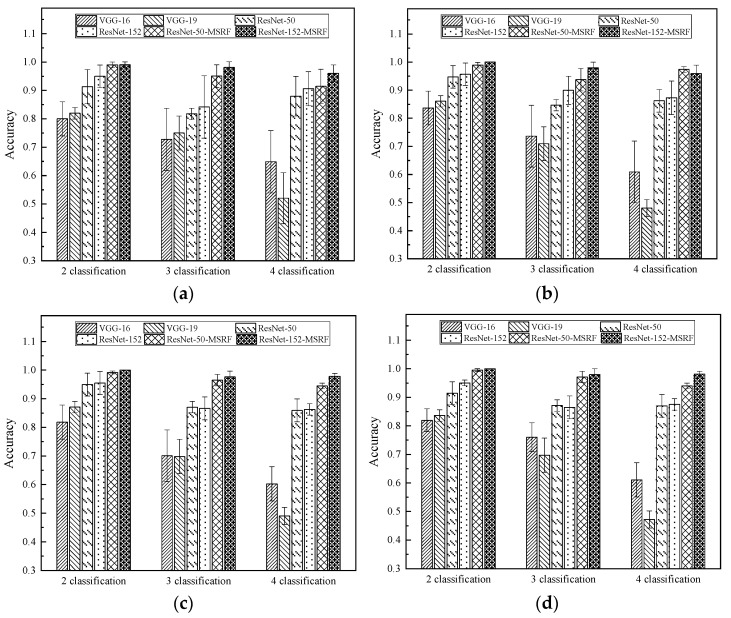
Comparison of the accuracy of six neural network models trained and validated under four different datasets, (**a**–**d**) represent different dataset contents. (**a**) A total of 3 classifications consists of normal, outer raceway fault and inner raceway fault; 2 classification consists of normal and outer raceway fault, (**b**) 3 classifications consists of normal, outer raceway fault and roller fault; 2 classifications consists of normal and inner raceway fault, (**c**) 3 classifications consists of normal, inner raceway fault and roller fault; 2 classifications consists of normal and roller fault, (**d**) 3 classifications consists of outer raceway fault, inner raceway fault and roller fault; 2 classifications consists of inner raceway fault and roller fault.

**Figure 9 sensors-22-01705-f009:**
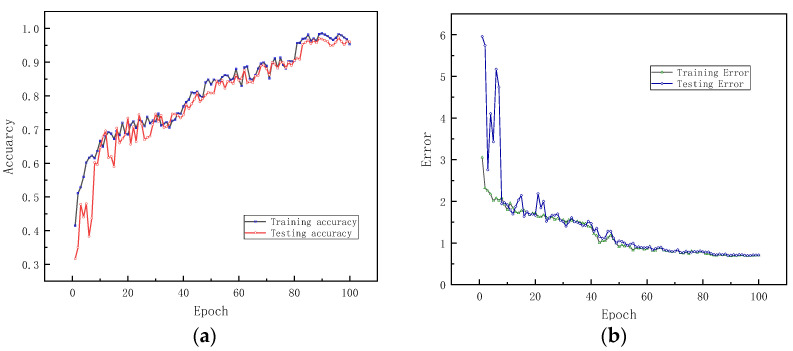
Accuracy and error comparison between ResNet-50-MSRF and ResNet-152-MSRF. (**a**) ResNet-152-MSRF training and testing accuracy, (**b**) ResNet-152-MSRF training and testing error, (**c**) ResNet-50-MSRF training and testing accuracy, (**d**) ResNet-50-MSRF training and testing error.

**Table 1 sensors-22-01705-t001:** Comparison of the number of computational parameters by multi-layer stacked convolutional kernels inserted into different networks.

	3 × 3 + 3 × 3	5 × 5	3 × 3 + 3 × 3 + 3 × 3	7 × 7
VGG-16	18,952,131	18,310,787	19,137,027	19,102,595
VGG-19	53,595,203	53,544,835	53,779,715	53,746,051
ResNet-50	762,691	714,703	1,234,093	1,197,084
ResNet-152	2,419,171	2,397,061	6,679,779	6,434,577

**Table 2 sensors-22-01705-t002:** Health conditions of bearings in the subway train axle box.

Category	Health Conditions	Sampling Points (per Series)	Series	Total Sampling Points	Rotation Speed (rpm)	Loads (KN)
1	Roller fault	122,581	16	1,961,296	1752	36/72
2	Inner raceway fault	125,049	16	2,000,784	1751	36/72
3	Outer raceway fault	122,514	16	1,960,224	1751	36/72
4	Normal	122,500	16	1,960,000	1750	36/72
5	Total	-	64	7,882,304	-	36/72

**Table 3 sensors-22-01705-t003:** Dataset size for each kind of health condition.

Fault Category	Roller Fault	Inner Raceway Fault	Outer Raceway Fault	Normal	Total
Sample size	5620	5742	5505	4245	21,112
Image size	128 × 128	128 × 128	128 × 128	128 × 128	128 × 128
Number of feature pixel points	92,078,080	94,076,928	90,193,920	69,550,080	345,899,008

**Table 4 sensors-22-01705-t004:** Architecture-related hyperparameters of the VGG-16, VGG-19, ResNet-50, ResNet-152, ResNet-50-MSRF, and ResNet-152-MSRF in the experiment.

Components	VGG-16	VGG-19	ResNet-50	ResNet-152	ResNet-50-MSRF	ResNet-152-MSRF
Input	128 × 128 × 3	128 × 128 × 3	128 × 128 × 3	128 × 128 × 3	128 × 128 × 3	128 × 128 × 3
Conv_2	13	16	50	152	50	152
Conv_kernel	(3 × 3, 1)	(3 × 3, 1)	(3 × 3, 2)	(3 × 3, 2)	(3 × 3 × 3, 2)	(3 × 3 × 3, 2)
Strides	1	1	1	1	1	1
BN	15	18	48	150	48	150
Activation function_1	ReLU	ReLU	ReLU	ReLU	ReLU	ReLU
RBU	0	0	1	1	1	1
Activation function_2	softmax	softmax	softmax	softmax	softmax	softmax
FC	3	3	2	2	2	2
Loss function	Category_crossentropy	Category_crossentropy	Category_crossentropy	Category_crossentropy	Category_crossentropy	Category_crossentropy
output	4	4	4	4	4	4
Optimizer	Adam	Adam	Adam	Adam	Adam	Adam
Lr	0.0001	0.0001	0.0001	0.0001	0.0001	0.0001
Dropout	0.5	0.5	0.5	0.5	0.5	0.5

**Table 5 sensors-22-01705-t005:** Average accuracy of the results in Figure 8 (including the lowest and highest accuracy).

Method	2 Classification Accuracy (%)	3 Classification Accuracy (%)	4 Classification Accuracy (%)
VGG-16	$81.85_{-1.85}^{+1.82}$	$73.13_{-3.06}^{+2.91}$	$61.79_{-1.53}^{+3.10}$
VGG-19	$84.68_{-2.09}^{+2.38}$	$71.37_{-1.66}^{+3.64}$	$49.09_{-1.88}^{+2.95}$
ResNet-50	$93.18_{-1.81}^{+1.75}$	$85.14_{-3.40}^{+1.96}$	$86.78_{-0.87}^{+1.22}$
ResNet-152	$95.30_{-0.30}^{+0.39}$	$86.81_{-2.63}^{+3.19}$	$87.92_{-1.68}^{+2.71}$
ResNet-50-MSRF	$99.15_{-0.22}^{+0.34}$	$95.61_{-1.86}^{+1.50}$	$94.32_{-2.85}^{+3.09}$
ResNet-152-MSRF	$99.77_{-0.70}^{+0.23}$	$97.93_{-0.31}^{+0.20}$	$96.99_{-1.04}^{+1.13}$

**Table 6 sensors-22-01705-t006:** Average calculation time of six different neural network models.

Method	Training/Testing Batch Size	Epochs	Total Calculation Time (s)	Calculation Time per Step (s)
VGG-16	20/10	100	5019	50.19
VGG-19	20/10	100	6467	64.67
ResNet-50	20/10	100	2536	25.36
ResNet-152	20/10	100	6974	69.74
ResNet-50-MSRF	20/10	100	2744	27.44
ResNet-152-MSRF	20/10	100	7363	73.63

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
