# Peer review of "Bearing Fault Reconstruction Diagnosis Method Based on ResNet-152 with Multi-Scale Stacked Receptive Field"

_sensors, 2022, doi:10.3390/s22051705_

Round 1

Reviewer 1 Report

In this work is proposed  a novel fault reconstruction characteristics classification method based on deep residual network with multi-scale stacked receptive field for rolling bearings of subway train axle box.

The proposal is interesting since is given a solution to real problems, however, there are some issues that have been attended.

1. The contribution of this proposal are mentioned at the end of Section 1 "introduction"can the authors mention the novelty of this work?

2. It is not clear how the setting parameters of the neural networks are defined. Some times the use of Genetic Algorithm (GA) can help to define such parameters, such as in https://doi.org/10.3390/s21175832, can you please discuss whether the use of GA may help to find optimum settings parameters for this proposal.

3. Please include future work in the conclusion section.

Author Response

Thank you for your correction! My response is attached.

Reviewer 2 Report

Dear Authors,

Please refer to my comments included in the attached file.

Author Response

(The authors gave the same response as above.)
